# Effects of GHRH Deficiency and GHRH Antagonism on Emotional Disorders in Mice

**DOI:** 10.3390/cells12222615

**Published:** 2023-11-12

**Authors:** Lucia Recinella, Maria Loreta Libero, Serena Veschi, Anna Piro, Guya Diletta Marconi, Francesca Diomede, Annalisa Chiavaroli, Giustino Orlando, Claudio Ferrante, Rosalba Florio, Alessia Lamolinara, Renzhi Cai, Wei Sha, Andrew V. Schally, Roberto Salvatori, Luigi Brunetti, Sheila Leone

**Affiliations:** 1Department of Pharmacy, “G. d’Annunzio” University of Chieti-Pescara, 66013 Chieti, Italy; lucia.recinella@unich.it (L.R.); veschi@unich.it (S.V.); anna.piro@unich.it (A.P.); annalisa.chiavaroli@unich.it (A.C.); giustino.orlando@unich.it (G.O.); claudio.ferrante@unich.it (C.F.); rosalba.florio@unich.it (R.F.); sheila.leone@unich.it (S.L.); 2Department of Cell Biology, Physiology and Immunology, University of Cordoba, 14014 Cordoba, Spain; 3Department of Innovative Technologies in Medicine & Dentistry, “G. d’Annunzio” University of Chieti-Pescara, 66013 Chieti, Italy; guya.marconi@unich.it (G.D.M.); francesca.diomede@unich.it (F.D.); 4Department of Neuroscience Imaging and Clinical Sciences, “G. d’Annunzio” University of Chieti-Pescara, 66013 Chieti, Italy; alessia.lamolinara@unich.it; 5Veterans Affairs Medical Center, Miami, FL 33125, USA; renzi.c@hotmail.com (R.C.); weisha17@gmail.com (W.S.); aschally@med.miami.edu (A.V.S.); 6Division of Medical/Oncology and Endocrinology, Department of Medicine, Miller School of Medicine, University of Miami, Miami, FL 33136, USA; 7Sylvester Comprehensive Cancer Center, Miami, FL 33136, USA; 8Department of Medicine, Division of Endocrinology, Diabetes and Metabolism, Johns Hopkins University School of Medicine, Baltimore, MD 21205, USA; salvator@jhmi.edu

**Keywords:** GHRH deficiency, GHRH antagonism, anxiety, depression, mood disorders

## Abstract

Growth hormone (GH)-releasing hormone (GHRH) has been suggested to play a crucial role in brain function. We aimed to further investigate the effects of a novel GHRH antagonist of the Miami (MIA) series, MIA-602, on emotional disorders and explore the relationships between the endocrine system and mood disorders. In this context, the effects induced by MIA-602 were also analyzed in comparison to vehicle-treated mice with GH deficiency due to generalized ablation of the GHRH gene (GHRH knock out (GHRHKO)). We show that the chronic subcutaneous administration of MIA-602 to wild type (+/+) mice, as well as generalized ablation of the GHRH gene, is associated with anxiolytic and antidepressant behavior. Moreover, immunohistochemical and Western blot analyses suggested an evident activation of Nrf2, HO1, and NQO1 in the prefrontal cortex of both +/+ mice treated with MIA-602 (+/+ MIA-602) and homozygous GHRHKO (−/− control) animals. Finally, we also found significantly decreased *COX-2*, *iNOS*, *NFkB*, and *TNF-α* gene expressions, as well as increased P-AKT and AKT levels in +/+ MIA-602 and −/− control animals compared to +/+ mice treated with vehicle (+/+ control). We hypothesize that the generalized ablation of the GHRH gene leads to a dysregulation of neural pathways, which is mimicked by GHRH antagonist treatment.

## 1. Introduction

Anxiety and depression represent the most common psychiatric illnesses of modern society [1]. In particular, there has been an increase in the incidence of both diseases during the COVID-19 pandemic period, probably due to direct consequences of COVID-19 illness or other correlated events, such as lockdowns and economic austerity [2]. Anxiety and depressive disorders may also occur in the setting of endocrine diseases [3]. Over recent years, different studies have investigated the relationships between emotional disorders and endocrine system [4,5,6,7,8,9,10,11,12]. Human studies indicate that acquired growth hormone (GH) deficiency may impair mood and memory, and these effects are addressed by GH replacement [13]. Conversely, animal studies demonstrate that somatostatin, which inhibits GH release, has anti-anxiety effects [6,14].

In this context, our research group focused on the behavioral effects of the novel GHRH antagonists of the Miami (MIA) series, MIA-690 and MIA-602, demonstrating that both peptides induce anxiolytic and antidepressant effects after chronic treatment in mice subjected to different behavioral tests. The beneficial effects induced by the analogs, in particular MIA-602, seem to be related to increased brain-derived neurotrophic factor (BDNF)/tropomycin receptor kinase B (TrkB) signaling, along with modulatory effects on the nuclear factor erythroid 2-related factor 2 (Nrf2)-linked inflammatory and oxidative status [15,16,17]. Accordingly, we previously observed that mice with GH deficiency (GHD) due to generalized ablation of the GHRH gene (GHRH knock out (GHRHKO)) show anxiolytic and antidepressant behavior [9,10]. In this context, generalized ablation of the GHRH gene in mice induced isolated GHD, showing a phenotype very similar to that displayed in other rodent models of GHD. In particular, GHRHKO mice showed marked growth retardation, with a body weight about 60% of normal animals [18]. Moreover, Sun and collaborators (2013) performed body composition studies showing an increase in adiposity in GHRHKO mice [19].

It is well known that various chronic diseases, such as emotional disorders, including anxiety and depression are linked to inflammation and oxidative stress. The activation of the inflammatory and oxidative stress response induces the release of inflammatory markers, as well as the mobilization of immune cells that can gain access to the brain [20,21]. Accordingly, an increase in pro-inflammatory mediators, including nuclear factor kappa-light-chain-enhancer of activated B cells (NF-kB), interleukin (IL)-1 and IL-6, as well as cyclooxygenase-2 (COX-2), was found in anxiety- and depression-related conditions [22,23,24].

The aim of the present work was to further investigate the potential effects of MIA-602 on emotional disorders and explore the brain pathways involved in its protective activity on mood disorders. In this context, the potential beneficial properties induced by MIA-602 were also analyzed in comparison to vehicle-treated GHRHKO mice.

## 2. Materials and Methods

### 2.1. Animals

Homozygous GHRHKO (−/−) male mice (5 weeks old, weight 10–12 g, *n* = 12) and wild type (C57/BL6, +/+) male mice (5 weeks old, weight 20–25 g, *n* = 24) were used in our study. −/− mice offspring were generated by mating heterozygous males and females, as previously reported [18]. Only male mice were used to avoid any possible effects of hormonal changes in adult female mice. The animals were housed in Plexiglas cages (2–4 animals per cage; 55 × 33 × 19 cm), with a 14/10 h light/dark cycle and ad libitum access to water and food [25,26]. Mice were fed with a standard rodent chow (Prolab RMH2500, PMI Nutrition International, Brentwood, MO, USA). Housing conditions and experimentation procedures were strictly in agreement with the European Community ethical regulations (EU Directive n. 26/2014) on the care of animals for scientific research, and were approved by the Italian Health Ministry (Project n. 885/2018-PR). Housing was maintained with minimal background noise and constant temperature (21 ± 2 °C; 55 ± 5% humidity).

### 2.2. In Vivo Studies

After a 2-week acclimation period, +/+ mice were randomized into two groups and treated daily for 4 weeks with subcutaneous administration of the GHRH antagonist MIA-602 (5 µg) (+/+ MIA-602) or vehicle solution [0.1% DMSO (Sigma, St. Louis, MO, USA) and 10% propylene glycol] (+/+ control). −/− mice were treated daily for 4 weeks with subcutaneous administration of vehicle solution (−/− control) [27]. Each test session was recorded using a video camera (SSC-DC378P, Biosite, Stockholm, Sweden) connected to a computer; a single video frame was acquired with a highly accurate, programmable, monochrome frame grabber board (Data TranslationTM, type DT3153), as previously described. The behavioral parameters were recorded at 4 weeks after the first treatment [15].

### 2.3. Behavioral Analysis

To evaluate anxiety-like behavior, each animal was placed in an open field box (40 × 40 × 31 cm) made from clear Plexiglas with a white laminated sheet of paper separated into 25 squares (8 × 8 cm each) covering the floor; the distance travelled (cm), and time spent into the center of the observation chamber (s) were recorded for 10 min [27].

The light–dark box test assessed bright-space related anxiety and consisted of two compartments (10 × 15 × 20 cm, each), a dark and a light one, separated by a wall pierced with an open door. The dark compartment had opaque black walls, while the light compartment was transparent to light. Mice were placed in the black compartment, and the time spent by the animal in the light compartment, as well as latency of first exit from the dark compartment, was recorded during a 10 min interval [28].

The elevated plus maze test consisted of two open arms and two closed arms that extended from a common central platform, elevated at a height of 45 cm above floor level, and mice were individually placed in the center of the maze facing an open arm. The time spent in open arms and the latency to first exit were recorded during a 10 min test period, as previously reported [11,29]. The evaluations were performed at 4 weeks of treatment.

The tail suspension test was used to evaluate despair behavior. Mice were individually suspended by the tail to a horizontal bar (at the height of 30 cm from floor) using adhesive tape. Immobility time was recorded during a 6 min period [9,10,11].

### 2.4. Hematoxylin-Eosin Staining/Light Microscopy Analysis and Immunohistochemistry

After euthanasia, brains were rapidly removed. Slices of the prefrontal cortex were dissected by using Paxinos and Watson Atlas [30]. Hematoxylin-eosin and immunohistochemical analyses in mouse prefrontal cortexes were performed as previously reported [31].

Briefly, mouse prefrontal cortexes were fixed in 10% phosphate-buffered formalin for 2.5 h. Each tissue block was dehydrated in a series of alcohol solutions of 50%, 70%, 96% and 99%, and then in Bioclear. Samples were then paraffin-embedded and cut into 7 μm-thick sections. Sections were de-waxed (Bioclear and alcohol in pro-gressively lower concentrations), rehydrated and processed for hematoxylin-eosin and for anti-Nrf2 immunohistochemical analysis. Primary antibody anti-Nfr2 (rabbit polyclonal, sc-722, Santa Cruz Biotechnology, Dallas, CA, USA) was applied for 2 h at room temperature and diluted to 1:200 in PBS (1×). The immunohistochemical reactions were revealed with the Rabbit specific HRP/DAB detection IHC kit (ab236469). The peroxidase reaction was developed using diaminobenzidine (DAB) chromogen, and nuclei were counterstained with hematoxylin. Lastly, sections were dehydrated, cleared with Bioclear and mounted in Eukitt mounting medium (Bio Optica, Milano, Italy). A negative control was established by omitting the primary antibody. Samples were then observed by means of LEICA DM 4000 B light microscopy (Leica Cambridge Ltd., Cambridge, UK) equipped with a Leica DFC 320 camera (Leica Cambridge Ltd.) for computerized images.

### 2.5. RNA Extraction, Reverse Transcription and Real-Time Reverse Transcription Polymerase Chain Reaction

Prefrontal cortexes were rapidly removed, and stored in RNAlater solution (Ambion, Austin, TX, USA) at −20 °C until further processing. The extraction of total RNA from prefrontal cortexes was performed by using TRI Reagent (Sigma–Aldrich, St. Louis, MO, USA), according to the manufacturer’s protocol. One microgram of total RNA extracted from each sample in a 20 μL reaction volume was reverse transcribed using a High-Capacity cDNA Reverse Transcription Kit (Thermo Fisher Scientific Inc., Monza, Italy). The gene expression of *COX-2*, inducible nitric oxide synthase (*iNOS*), *NF-kB* and tumor necrosis factor (*TNF*)*-α* in prefrontal cortex specimens was determined by means of quantitative real-time polymerase chain reaction (PCR), using TaqMan probe-based chemistry (Thermo Fisher Scientific Inc., Monza, Italy), as previously reported [16,32]. PCR primers and TaqMan probes were purchased from Thermo Fisher Scientific Inc. (Assays-on-Demand Gene Expression Products, Mm00478374_m1 for *COX-2*, Mm00443258_m1 for *TNF-α*, Mm00476361_m1 for *NF-kB*, Mm00440502_m1 for *iNOS*, and Mm00607939_s1 for *β-actin*). *β-actin* was used as the housekeeping gene.

### 2.6. Western Blot Analysis

Western blotting in prefrontal cortex tissues was performed as described previously [33]. After the collection of cortex tissues, samples were homogenized by using Ultra-Turrax homogenizer (IKA-Werke, Staufen, Germany) in RIPA buffer added with 1 mM PMSF (Phenylmethanesulfonyl Fluoride) and protease and phosphatase inhibitor cocktails (Sigma). Lysed samples were sonicated and centrifuged at 15,000 rpm (4 °C for 20 min). Quantification of protein concentrations was performed using the BCA Protein Assay (Thermo Scientific, Rockford, IL, USA). In this process, 40 μg of protein lysates was subjected to electrophoresis followed by immunoblotting [2]. The nitrocellulose membranes were blocked in 5% nonfat dry milk and incubated overnight at 4 °C with the appropriate primary antibodies. Anti-TrkB and anti-HO-1 rabbit monoclonal antibodies were purchased from Cell Signaling Technology, Inc. (Beverly, MA, USA). Goat polyclonal anti-NQO1, anti-P-AKT, anti-AKT, anti-PI3K antibody were obtained from Abcam (Cambridge, UK). The membranes were then incubated with either anti-rabbit or anti-mouse HRP-conjugated secondary antibodies (Cell Signaling Technology, Beverly, MA, USA). The blots were revealed with the Westar ηC Ultra 2.0 chemiluminescence substrate (Cyanagen, Bologna, Italy). GAPDH was used as a loading control (Santa Cruz Biotechnology).

### 2.7. Statistical Analysis

Statistical analysis was performed using GraphPad Prism version 5.01 for Windows (GraphPad Software, San Diego, CA, USA). All data were collected from each of the animals used in the experimental procedure and means ± SEM were determined for each experimental group and analyzed by means of 2-way analysis of variance (ANOVA) followed by Bonferroni post hoc tests. Statistical significance was accepted at *p* < 0.05. As regards gene expression analysis, the comparative 2^−ΔΔCt^ method was used to quantify the relative abundance of mRNA and then to determine the relative changes in individual gene expression (relative quantification) [34]. Finally, as regards the animals randomized for each experimental group, the number was calculated on the basis of the ‘Resource Equation’ N = (E + T)/T (10 ≤ E ≤ 20) [35].

## 3. Results

### 3.1. Anxiety-Related and Depression-like Behavior

To investigate anxiety-related behavior, the open field test, the light–dark box test and elevated plus maze test were used. In the open field test, +/+ MIA-602 and −/− control mice traveled a greater distance (Figure 1A, *p* < 0.005 (for −/− control and +/+ MIA-602 mice)) and spent significantly more time in the central zone (Figure 1B, *p* < 0.005 (for +/+ MIA-602 mice) and *p* < 0.01 (for −/− control mice)), compared to the +/+ control group. In addition, +/+ MIA-602 mice spent significantly more time in the central zone compared to −/− control mice (Figure 1B, *p* < 0.05).

In the light–dark box and elevated plus maze tests, the time spent in the light area and open arms, respectively, was significantly higher in +/+ MIA-602 mice and −/− control mice (Figure 1C,E, *p* < 0.005 (for +/+ MIA-602 mice) and *p* < 0.01 (for −/− control mice)) compared to the +/+ control group. We found that both the −/− control and +/+ MIA-602 mice showed decreased latencies to emerge from the dark compartment (Figure 1D, *p* < 0.005 (for +/+ MIA-602 mice) and *p* < 0.01 (for −/− control mice)) and from the central zone in the elevated plus maze (Figure 1F, *p* < 0.01 (for +/+ MIA-602 and −/− control mice)) compared to the +/+ control group. In particular, the time spent in the light area and open arms was higher in +/+ MIA-602 compared to −/− control mice (Figure 1C,E, *p* < 0.05). Collectively, our findings show that the reduction in anxiety-related behavior was more marked in +/+ MIA-602 with respect to −/− control mice.

Additionally, we evaluated behavioral despair via the tail suspension test. −/− control and +/+ MIA-602 mice showed a significant reduction in total immobility (Figure 1G, *p* < 0.001 (for +/+ MIA-602 mice) and *p* < 0.005 (for −/− control mice)), compared to the +/+ control group. In the tail suspension test, total immobility was lower in +/+ MIA-602 than in −/− control mice (Figure 1G, *p* < 0.05).

### 3.2. Hematoxylin-Eosin Staining and Immunohistochemical Analysis of Nrf2 in Mice Prefrontal Cortex

Morphological features and the detection of Nrf2 in the prefrontal cortex were analyzed by means of hematoxylin-eosin (H&E) staining and immunohistochemistry, respectively. H&E-stained sections of (a) +/+ MIA-602, (b) −/− vehicle-treated (−/− control), and (c) +/+ vehicle-treated (+/+ control) showed a normal histological structure of the prefrontal cortex (Figure 2A). Our results also show an increase in the immunoreactivity for Nrf2 in +/+ MIA-602 and −/− control animals with respect to +/+ control mice (Figure 2A, *p* < 0.01 (for +/+ MIA-602) and *p* < 0.05 (for −/− control)). In particular, the increase in immunoreactivity for Nrf2 was higher in +/+ MIA-602 compared to −/− control animals (Figure 2A, *p* < 0.05).

### 3.3. Heme Oxygenase-1, NAD(P)H Quinone Oxidoreductase 1, Tropomycin Receptor Kinase B, and Brain-Derived Neurotrophic Factor Evaluation in Prefrontal Cortex

The results from Western blot analysis and real-time polymerase chain-reaction (PCR) revealed that heme oxygenase-1 (HO1), NAD(P)H quinone oxidoreductase 1 (NQO1), TrkB and BDNF (Figure 2B–D; Appendix A) are expressed in the prefrontal cortex. In particular, we found that HO1 and NQO1 protein levels in the prefrontal cortex were increased in +/+ MIA-602 and −/− control animals compared to the +/+ control group (Figure 2B,C, *p* < 0.01 (for +/+ MIA-602 and −/− control mice, panel B); *p* < 0.05 (for +/+ MIA-602, panel C) and *p* < 0.01 (for −/− control mice, panel C)). The increase in NQO1 protein levels was higher in −/− control mice compared to +/+ MIA-602 animals (Figure 2C, *p* < 0.05).

Our findings also show that TrKB (full length form (140 kDa) and truncated form (90~95 kDa)) protein and BDNF gene expression levels were increased in +/+ MIA-602 compared to −/− control and +/+ control mice (Figure 2D, *p* < 0.005 (for +/+ MIA-602), panel E, *p* < 0.05 (for +/+ MIA-602)) (Appendix A).

### 3.4. Cyclooxygenase-2, Inducible Nitric Oxide Synthase, Nuclear Factor Kappa-Light-Chain-Enhancer of Activated B Cells and Tumor Necrosis Factor-α Gene Expression Determination in the Prefrontal Cortex

Real-time reverse transcription PCR analysis revealed a significant decrease in *COX-2* (A), *iNOS* (B), *NF-kB* (C) and *TNF-α* (D) gene expression in +/+ MIA-602 and −/− control animals with respect to the +/+ control group in the prefrontal cortex (Figure 3A–D, *p* < 0.05).

### 3.5. Phosphorylated AKT, Serine/Threonine Kinase, and Phosphoinositide 3-Kinase Protein Expression Levels

In order to investigate the potential molecular mechanism of the reduction in anxiety- and antidepressive-like-related behavior of +/+ MIA-602 and −/− control animals, we studied the serine/threonine kinase (AKT), phosphorylated AKT (P-AKT), and phosphoinositide 3-kinase (PI3K) protein levels in the prefrontal cortex. In this context, Palumbo and collaborators (2021) showed that AKT deficiency could induce anxiety and depressive behavior [36]. Moreover, the role of PI3K in depression and anxiety is widely known [36,37,38]. In agreement with this, the PI3K signaling pathway was found to be involved in the central stress-induced modulation of synaptic plasticity [39].

We found that +/+ MIA-602 and −/− control animals showed higher P-AKT and AKT levels in the prefrontal cortex with respect to the +/+ control group (Figure 4A, *p* < 0.01 (for +/+ MIA-602 mice) and *p* < 0.05 (for −/− control mice)). In particular, treatment with MIA-602 in +/+ mice induced higher stimulation of both the P-AKT and AKT levels with respect to −/− control mice (Figure 4A, *p* < 0.05)) (Appendix A). On the other hand, PI3K protein expression levels in the prefrontal cortex were not modified in +/+ mice after treatment with MIA-602 or in −/− control mice with respect to the +/+ control group (Figure 4B) (Appendix A).

## 4. Discussion

GHRHKO animals represent a model of GH deficiency with normal pituitary function. In particular, GHRHKO mice displayed decreased pituitary GH mRNA and protein levels, as well as decreased serum insulin-like growth factor 1 (IGF-1) levels and liver IGF-1 gene expression compared to control animals [18]. Moreover, Sun and collaborators (2013) reported a reduction in plasma glucose levels as well as increased insulin sensitivity in the same animal model [19].

We previously demonstrated that GHRH deficiency induces a reduction in anxiety- and depression-related behavior in male mice, as well as decreased norepinephrine levels in the striatum area with respect to control animals [9,11]. Moreover, GHRH antagonists, such as MZ-4-71, as well as MIA-690, exert anxiolytic and anti-depressant activities [17,27,40].

Accordingly, in the present study, we show that the generalized ablation of the GHRH gene, as well as the chronic subcutaneous administration of the GHRH antagonist MIA-602, is associated with anxiolytic and antidepressant behavior in mice (Figure 1), confirming the crucial role of GHRH in animal models of mood disorders.

In cerebral areas, including the prefrontal cortex, Nrf2, a well-known redox-sensitive transcription factor involved in the cellular defense against oxidative stress, and BDNF, a growth factor which is involved in multiple aspects of synaptogenesis, have been hypothesized to represent therapeutic targets for mood disorders [41,42,43]. In particular, BDNF and its receptor, TrkB, were found to be increased after chronic treatment with fluoxetine, a selective serotonin reuptake inhibitor (SSRI) [44,45,46,47]. Based on these observations, in this study, we aimed to measure the expression of Nrf2 and its target genes (HO1 and NQO1), as well as the potential activation of BDNF and its receptor, TrkB, in the prefrontal cortex of +/+ MIA-602 and −/− control mice. In our experimental protocol, immunohistochemical and Western blot analyses suggested an evident activation of Nrf2, HO1, and NQO1 in the prefrontal cortex of both +/+ MIA-602 and −/− control animals. However, BDNF and TrkB expression levels were significantly higher only in the prefrontal cortex of +/+ MIA-602 mice (Figure 2). The reduction in the abundance of anti-inflammatory and antioxidant markers (Figure 3) further confirms the activation of the NrF2/HO-1/NQO1 pathway previously observed in the prefrontal cortex of both groups of animals [27].

To further evaluate the potential molecular mechanisms involved in the antianxiety and antidepressive behavior of +/+ MIA-602 and −/− control animals, we measured the protein expression levels of AKT serine/threonine kinase, phosphorylated AKT (P-AKT), and PI3K in the prefrontal cortex using Western blotting analysis. In this context, Matsuda and collaborators showed that the PI3K/AKT pathway plays a key role in cell growth, survival and proliferation, and is critically involved in synaptic transmission and nerve plasticity [48]. In particular, it is well known that the PI3K/AKT signaling pathway plays a pivotal role in the regulation of neurotrophy and neuroinflammation, thus suggesting its involvement in the etiology of mood disorders [49]. Accordingly, Leibrock and collaborators (2013) found that Akt2 knockout (Akt2 −/−) mice, besides showing cognitive impairment, displayed an anxiety- and depressive-like phenotype [50]. In addition, AKT activity is reduced in some brain regions of patients with a diagnosis of major depression [51]. P-AKT levels were also found to be decreased in an animal model of depression [52].

We found a significant increase in the levels of P-AKT and AKT, but not those of PI3K, in both +/+ MIA-602 and −/− control animals with respect to the +/+ control group (Figure 4). In particular, P-AKT and AKT levels were higher in +/+ MIA-602 mice with respect to −/− control mice.

More recently, increased AKT phosphorylation was also shown to be involved in the rapid antidepressant-like effects of various drugs in several areas of the central nervous system, including the prefrontal cortex and hippocampus [53,54,55]. Interestingly, downregulation of the PI3K-AKT signaling pathway was shown to induce phosphorylation of NF-κB, which is involved in modulating the expression of various genes as well as the intracellular transmission of different signals. Moreover, the activation of the NF-κB signaling pathway is involved in depression development [56]. Accordingly, elevated levels of pro-inflammatory mediators in mood disorders have been reported in various studies [22,23,24].

In this context, NF-κB has been hypothesized to stimulate the release of inflammatory mediators, including TNF-α, interleukin (IL)-1β, and IL-6, as well as promoting neuroinflammatory processes, finally leading to depression [57]. Accordingly, Zhou and collaborators (2022) suggested that total saikosaponins (TSS) may alleviate depression-like behaviors induced by chronic unpredictable mild stress by modulating the PI3K/AKT/NF-κB signaling axis [58].

The two main limitations of our study are that our evaluations were not performed in female mice, and that we did not study emotional behavior in other animal models with a different etiology of GHD.

## 5. Conclusions

In summary, our data show that the molecular and behavioral effects induced by GHRH antagonism after MIA-602 treatment in +/+ animals are very similar to those observed in GHRHKO −/− control mice. In addition to showing anxiolytic and antidepressant behavior, −/− mice are characterized by an increased lifespan [19,59]. Similarly, MIA-690, a GHRH antagonist, induced beneficial effects in different models of Alzheimer’s disease, showing anti-oxidative and neuro-protective effects [60]. In addition, we previously demonstrated that both −/− mice and +/+ animals treated with GHRH antagonists showed hyperphagia [17,61]. Therefore, we can hypothesize that the generalized ablation of the GHRH gene leads to a dysregulation of neural pathways, which is mimicked by GHRH antagonist treatment. This could involve a direct role of GHRH receptors, or be mediated by the reduced production of IGF-I and GH.

## 6. Patents

A.V.S. and R.C. are listed as co-inventors on patents for GHRH antagonists, assigned to the University of Miami, Miami, FL, and the Veterans Affairs Medical Center, Miami, FL.

## Figures and Tables

**Figure 1 cells-12-02615-f001:**
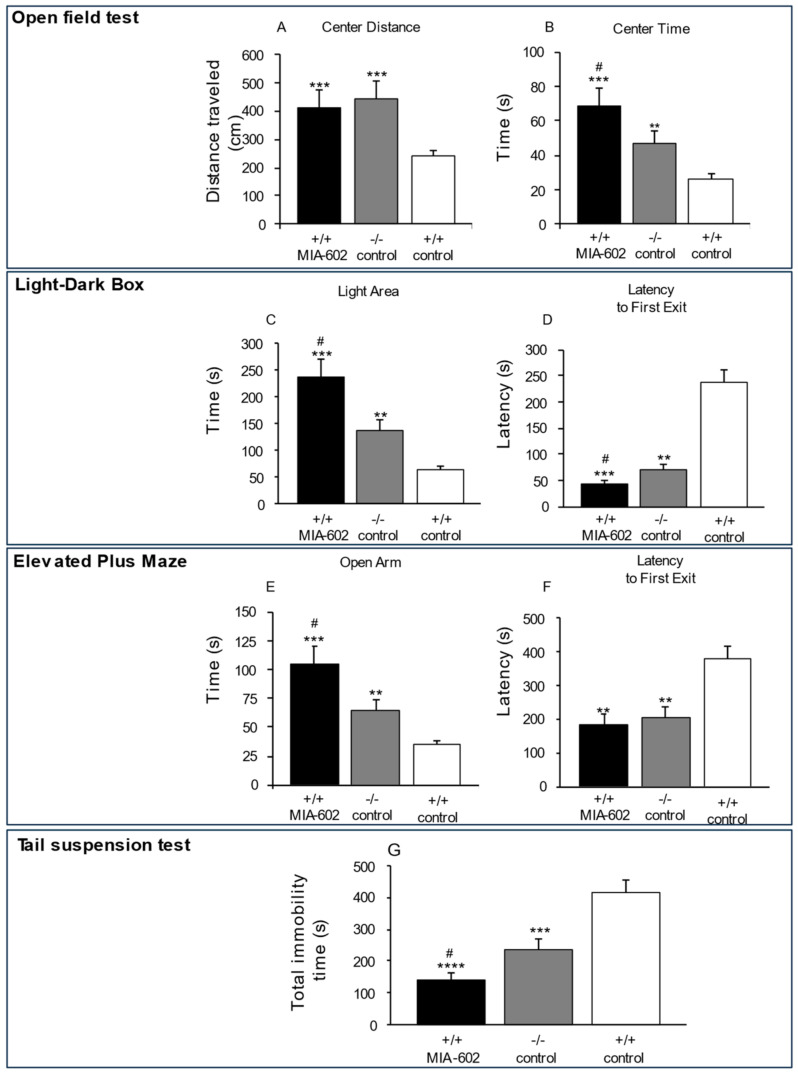
(**A**–**G**) Analysis of anxiety-related behavior and behavioral despair in +/+ mice treated with MIA-602 (5 μg) (+/+ MIA-602) and −/− mice treated with vehicle (−/− control) (*n* = 12 for each group of treatment). Compared to vehicle-treated +/+ mice (+/+ control), a reduction in anxiety-like behavior and behavioral despair in the open field test, the light–dark box test, the elevated plus maze test and the tail suspension test was found in +/+ mice MIA-602 and −/− control mice. The reduction in anxiety-like behavior and behavioral despair was higher in +/+ MIA-602 with respect to −/− control mice. Data are expressed as means ± SEM and analyzed by analysis of variance (ANOVA) followed by Bonferroni post hoc tests. ANOVA, ** *p* < 0.01, *** *p* < 0.005, **** *p* < 0.001 vs. +/+ control; # *p* < 0.05 vs. −/− control.

**Figure 2 cells-12-02615-f002:**
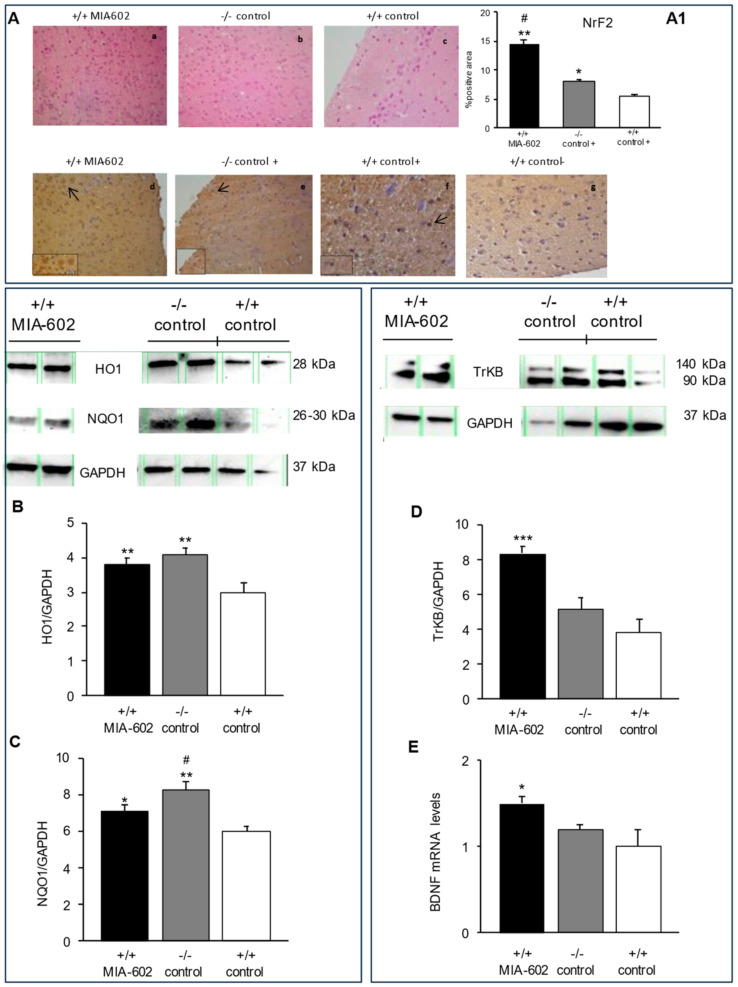
Hematoxylin-eosin staining and immunohistochemical analysis of Nrf2 expression in the mouse prefrontal cortex (*n* = 6 for each group of treatment). (**A**) Hematoxylin-eosin staining: (**a**) MIA-602-treated +/+ mice (+/+ MIA-602); (**b**) −/− vehicle-treated mice (−/− control); (**c**) +/+ vehicle-treated mice (+/+ control). Scale bar: 100 µm, magnification 20×. (**A**) Immunohistochemical detection of Nrf2 expression: (**d**) +/+ MIA-602; (**e**) −/− control +; (**f**) +/+ control +; (**g**) negative control (+/+ control−). Images show Nrf2 nuclear staining; arrows indicate Nfr2-positive area. Scale bar: 100 µm, magnification 20×. (**A1**) Graphic representation of the percentage of Nrf2-positive area (±SD); densitometric analysis determined by means of direct visual counting of ten fields for each of the three slides per sample. * *p* < 0.05, ** *p* < 0.01, *** *p* < 0.005 vs. +/+ control+, # *p* < 0.05 vs. −/− control +. Protein expression for HO1 (**B**), NQO1 (**C**), and TrkB (full length form (140 kDa) and truncated form (90~95 kDa)) (**D**) and relative quantification of gene expression of BDNF (**E**) in the prefrontal cortex (*n* = 6 for each group of treatment). As for gene expression, data were calculated using the 2^−ΔΔCt^ method, normalized to β-actin mRNA levels, and expressed relative to the control (calibrator sample, defined as 1.00). Data are expressed as means ± SEM. * *p* < 0.05 vs. +/+ control+.

**Figure 3 cells-12-02615-f003:**
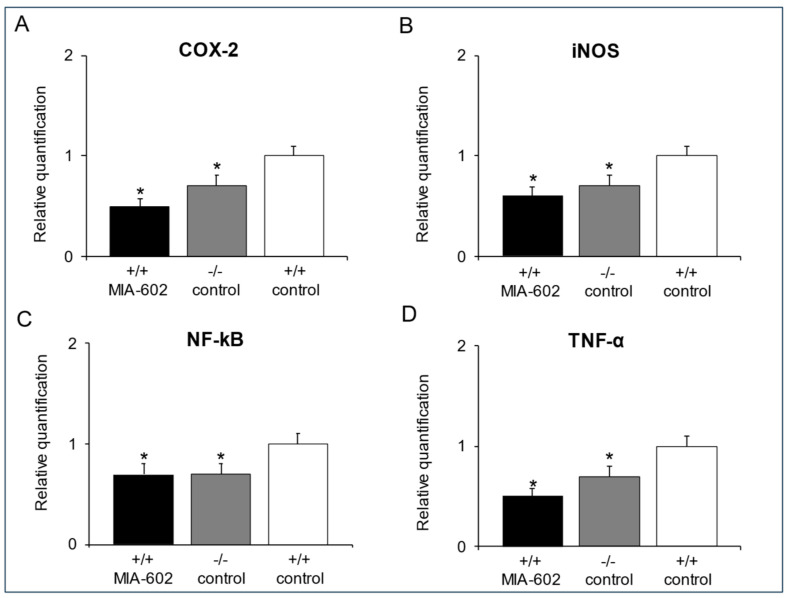
(**A–D**) Inhibitory effects of MIA-602 (5µg) on *COX-2*, *iNOS*, *NF-kB* and *TNF-α* gene expression in the prefrontal cortex (*n* = 6 for each group of treatment). Data were calculated using the 2^−ΔΔCt^ method, being normalized to β-actin mRNA levels and then expressed relative to the control (calibrator sample, defined as 1.00). Data are expressed as means ± S.E.M. * *p* < 0.05 vs. +/+ control.

**Figure 4 cells-12-02615-f004:**
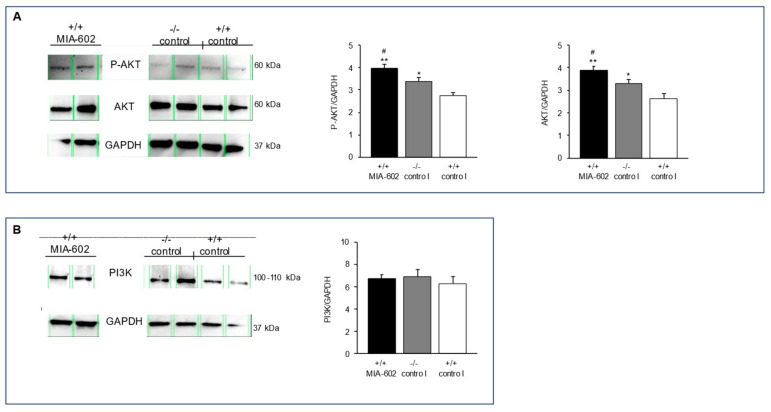
Western blot of P-AKT (60 kDa) (**A**), AKT (60 kDa) (**A**), and PI3K (100–110 kDa) (**B**) proteins, in which GAPDH (37 kDa) was used as a loading control, in the prefrontal cortex of +/+ mice treated with MIA-602 (5 μg) (+/+ MIA-602) and −/− mice treated with vehicle (−/− control) (*n* = 6 for each group of treatment). Data are expressed as means ± S.E.M. (*n* = 8 for each group); * *p* < 0.05 and ** *p* < 0.01 vs. +/+ control; # *p* < 0.05 vs. −/− control.

## Data Availability

The datasets used and/or analyzed during the current study are available from the corresponding author on reasonable request.

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
