# Peer review of "Effects of GHRH Deficiency and GHRH Antagonism on Emotional Disorders in Mice"

_cells, 2023, doi:10.3390/cells12222615_

Round 1

Reviewer 1 Report

Comments and Suggestions for Authors

The study from Recinella et al., evaluating the effects of the antagonist MIA-602 on emotional disorders, represents a continuation of work previously carried out by their research group. One of the goals of this study is to explore the relationship between the endocrine system and mood disorders. This study only uses behavioral assessments and does not provide any endocrine data, not even a description of the phenotype of GHRH-deficient animals. The aims of the study need to be adjusted accordingly. I also suggest a revision of the introduction section, providing details of the animal model phenotype so the correlation with the endocrine system can be explored in the discussion section.

 I have additional minor comments:

 Please use abbreviations the first time the terms are mentioned in the text; preferably, these descriptions should not be in the subtitle of the results topics.

 page 5, lines 172-173, the text "Immunohistochemical examination revealed positive immunostaining for Nrf2 expression in (d) +/+ MIA-602 and (e) 173 -/- control + compared to (f) +/+ control +" is unnecessary since the correct description of the result is in the following sentence (lines 174-176).

 The quality of Figure 2 is very poor. The photomicrographs in Fig 2A are out of focus or blurred, and the bar graph axes cannot be read. Please improve Fig 2A accordingly. Also, remove the numbering from the scale bar in the photomicrographs in Figure 2A; this is unreadable. Please insert the bar with the appropriate size and quote the corresponding value in the legend.

 In the discussion section, page 9, lines 302-303, it is important to recognize that the lack of data regarding female mice is also a limitation of this study. Please make a paragraph contextualizing what is known, or not, about how sex differences contribute to the role of GHRH effects on emotional disorders, and further recognize this additional limitation. 

Author Response

Dear Editor,

thanks for the questions posed by the Reviewers.

We have revised our manuscript [Manuscript ID: cells-2665071, Title: Effects of GHRH deficiency and GHRH antagonism on emotional disorders in mice], and the questions posed by the Reviewers have been responded point by point. The changes have been highlighted in yellow in the revised version of the manuscript.

Reviewer 1:

Comments and Suggestions for Authors

The study from Recinella et al., evaluating the effects of the antagonist MIA-602 on emotional disorders, represents a continuation of work previously carried out by their research group. One of the goals of this study is to explore the relationship between the endocrine system and mood disorders. This study only uses behavioral assessments and does not provide any endocrine data, not even a description of the phenotype of GHRH-deficient animals. The aims of the study need to be adjusted accordingly. I also suggest a revision of the introduction section, providing details of the animal model phenotype so the correlation with the endocrine system can be explored in the discussion section.

Response to Reviewer: We thank the Reviewer for the comment. In the Revised Manuscript, we have provided details of the animal model phenotype (introduction section), as well as the correlation with the endocrine system (discussion section).

 I have additional minor comments:

 Please use abbreviations the first time the terms are mentioned in the text; preferably, these descriptions should not be in the subtitle of the results topics.

Response to Reviewer: We thank the Reviewer for the comment. In the Revised Manuscript we performed the kindly suggested correction. 

 page 5, lines 172-173, the text "Immunohistochemical examination revealed positive immunostaining for Nrf2 expression in (d) +/+ MIA-602 and (e) 173 -/- control + compared to (f) +/+ control +" is unnecessary since the correct description of the result is in the following sentence (lines 174-176).

Response to Reviewer: We thank the Reviewer for the comment. In the Revised Manuscript we performed the kindly suggested correction. 

 The quality of Figure 2 is very poor. The photomicrographs in Fig 2A are out of focus or blurred, and the bar graph axes cannot be read. Please improve Fig 2A accordingly. Also, remove the numbering from the scale bar in the photomicrographs in Figure 2A; this is unreadable. Please insert the bar with the appropriate size and quote the corresponding value in the legend.

Response to Reviewer: We thank the Reviewer for the suggestion. In Supplementary Materials, we have inserted photomicrographs in Figure 2A (Fig. S1) with higher resolution. In the Revised Manuscript, (Fig. 2A) we removed the numbering from the scale bar in the photomicrographs and quoted the corresponding value in the legend. We also improved the reading of the bar graph axes.

 In the discussion section, page 9, lines 302-303, it is important to recognize that the lack of data regarding female mice is also a limitation of this study. Please make a paragraph contextualizing what is known, or not, about how sex differences contribute to the role of GHRH effects on emotional disorders, and further recognize this additional limitation. 

Response to Reviewer: We thank the Reviewer for the comment. In the Revised Manuscript, we regognized the lack of data regarding female mice as a main limitation of our study. Actually, no study is avalaible investigating the possible involvement of sex differences on GHRH effects on emotional disorders.

Reviewer 2 Report

Comments and Suggestions for Authors

This paper investigated the effects of a GHRH antagonist, MIA-602 by subcutaneous injection using wild and GHRH gene knockout mice. Evaluation was performed using various parameters such as behavioral, biochemical and molecular levels. Data themselves are of interest. However, there are several concerns that the authors should address before publication.

1.    Introduction seems to be insufficient. Readers can understand why the authors investigated, for example, COX-2 gene after reading the Discussion. It is better to add summary of various parameters associated with anxiety and depression.

2.    Figure 2A are too small. It is difficult to judge which are nuclei and which are immuno-positive structures.

3.    In western blotting, GAPDH was used as a loading control. Then, GAPDH immunoreactive (explanation of this method is insufficient) bands were expected to show same density. In addition, in figure legends, GAPDH molecular weight is 36 kDa, but labeling in photographs is 37 kDa.

Minor points

1.    P. 6, L. 192: “shown (Fig. 2)” does not make sense.

2.    P. 9, L. 273: Add reference(s) in the end of this sentence, if present.

Author Response

Dear Editor,

thanks for the questions posed by the Reviewers.

We have revised our manuscript [Manuscript ID: cells-2665071, Title: Effects of GHRH deficiency and GHRH antagonism on emotional disorders in mice], and the questions posed by the Reviewers have been responded point by point. The changes have been highlighted in yellow in the revised version of the manuscript.

Reviewer 2:

Comments and Suggestions for Authors

This paper investigated the effects of a GHRH antagonist, MIA-602 by subcutaneous injection using wild and GHRH gene knockout mice. Evaluation was performed using various parameters such as behavioral, biochemical and molecular levels. Data themselves are of interest. However, there are several concerns that the authors should address before publication.

  1. Introduction seems to be insufficient. Readers can understand why the authors investigated, for example, COX-2 gene after reading the Discussion. It is better to add summary of various parameters associated with anxiety and depression.

Response to Reviewer: We thank the Reviewer for the comment. In the Revised Manuscript, we expanded Introduction section, by describing the role of inflammation and oxidative stress in anxiety- and depression-related conditions.

  1. Figure 2A are too small. It is difficult to judge which are nuclei and which are immuno-positive structures.

Response to Reviewer: We thank the Reviewer for the suggestion. In Supplementary Materials, we have inserted photomicrographs in Figure 2A (Fig. S1) with higher resolution.

  1. In western blotting, GAPDH was used as a loading control. Then, GAPDH immunoreactive (explanation of this method is insufficient) bands were expected to show same density. In addition, in figure legends, GAPDH molecular weight is 36 kDa, but labeling in photographs is 37 kDa.

 Response to Reviewer: We thank the Reviewer for the comment. In Supplementary Materials, we detailed methods used for Western blot analysis. In addition, in figure legends, we corrected GAPDH molecular weight.

Minor points

  1. P. 6, L. 192: “shown (Fig. 2)” does not make sense.

Response to Reviewer: We thank the Reviewer for the comment. We performed the kindly suggested corrected.

  1. P. 9, L. 273: Add reference(s) in the end of this sentence, if present.

Response to Reviewer: We thank the Reviewer for the comment. We performed the kindly suggested corrected.